# Distantly related *Alteromonas* bacteriophages share tail fibers exhibiting properties of transient chaperone caps

Rafael Gonzalez-Serrano [1,5], Riccardo Rosselli [2,6], Juan J. Roda-Garcia[1], Ana-Belen Martin-Cuadrado[3], Francisco Rodriguez-Valera [1] & Matthew Dunne [4] ✉

The host recognition modules encoding the injection machinery and receptor binding proteins (RBPs) of bacteriophages are predisposed to mutation and recombination to maintain infectivity towards co-evolving bacterial hosts. In this study, we reveal how *Alteromonas mediterranea* schitovirus A5 shares its host recognition module, including tail fiber and cognate chaperone, with phages from distantly related families including *Alteromonas* myovirus V22. While the V22 chaperone is essential for producing active tail fibers, here we demonstrate production of functional A5 tail fibers regardless of chaperone co-expression. AlphaFold-generated models of tail fiber and chaperone pairs from phages A5, V22, and other *Alteromonas* phages reveal how amino acid insertions within both A5-like proteins results in a knob domain duplication in the tail fiber and a chaperone β-hairpin "tentacle" extension. These structural modifications are linked to differences in chaperone dependency between the A5 and V22 tail fibers. Structural similarity between the chaperones and intramolecular chaperone domains of other phage RBPs suggests an additional function of these chaperones as transient fiber "caps". Finally, our identification of homologous host recognition modules from morphologically distinct phages implies that horizontal gene transfer and recombination events between unrelated phages may be a more common process than previously thought among *Caudoviricetes* phages.

Host recognition modules of bacterial viruses (phages) encode the receptor binding proteins (RBPs) responsible for recognition and attachment to specific bacterial hosts and the tail infection apparatus (e.g., distal tail and/or baseplate components) responsible for puncturing bacterial cells and mediating viral genome translocation[1–4]. Typically identified as tail fibers or tailspikes (TSPs), RBPs are structurally diverse protein complexes that recognize exposed bacterial cell wall structures of saccharidic (e.g., lipopolysaccharide (LPS), teichoic acids, and capsular polysaccharides) and/or proteinaceous (e.g., outer membrane proteins, pili, and flagella) composition[1,4–6].

The high degree of compositional variation of saccharide-based phage receptors partly explains the commonly narrow host range

[1]Evolutionary Genomics Group, Universidad Miguel Hernández, San Juan de Alicante, Spain. [2]Research & Development Department, LABAQUA S.A. Las Atalayas, Alicante, Spain. [3]Department of Physiology, Genetics and Microbiology, University of Alicante, Alicante, Spain. [4]Institute of Food, Nutrition and Health, ETH Zurich, Switzerland. [5]Present address: Centro de Biología Molecular Severo Ochoa, CBMSO-CSIC, Madrid, Spain. [6]Present address: Department of Physiology, Genetics and Microbiology, University of Alicante, Alicante, Spain. ✉e-mail: matthew.dunne@hest.ethz.ch

observed for phages that solely use these structures for host recognition[7,8]. In the bacteria, the saccharidic composition of these structures is determined by hypervariable glycosylation genomic islands that are subject to frequent exchange between close or distant taxa of bacteria through horizontal gene transfer (HGT)[9–11]. To compensate for the high degree of receptor plasticity, phages must also adapt their RBPs to recognize changes in receptor composition within an equivalent evolutionary timeframe; something that cannot depend on random mutation alone. Subsequently, phages have evolved impressive counterstrategies for host range modification[1], from the use of hypermutable polyG tracts to provide phase variable RBP expression[12], to encoding reverse transcriptases that mediate RBP mutation[13], or by incorporating convertible tail fiber and chaperone pairs within reversible genome segments[14]. However, from a broader evolutionary perspective, and evidenced by the mosaicism observed across all phage genomes, phage evolution and host range adaptation appear to be principally driven by HGT and homologous recombination events between phages[15–18]. HGT between phage genomes requires the concurrent presence of two or more different phages within a host bacterium, for example, co-infecting virulent phages or virulent phages infecting a bacterial lysogen carrying prophage(s)[19]. And thus, it is unclear if genetic exchange between phages operates as widely or frequently as occurs in their bacterial hosts, and how this varies depending on the lifestyle, genomic composition, and target species of different phages[20].

*Alteromonas* phage V22 was previously isolated and identified as a member of a new *Myoalterovirus* genus within the *Myoviridae* family that employs chaperone-dependent tail fibers[21]. While the chaperone (gp27) was essential for recombinant production of functional tail fibers, gp27 was shown to remain weakly associated with the mature fiber after production, suggesting an alternative role besides assisting with fiber maturation. Interestingly, other tail fiber chaperones have been shown to remain attached to their tail fibers after production[22,23] and have also been proposed to play alternative roles beyond their conventional chaperone function. For example, the tail fiber assembly (Tfa) protein of *Escherichia coli* phage Mu remains bound to the tip of the tail fiber and has been suggested to interact with the same lipopolysaccharide receptor as the tail fiber[24].

In contrast to phage V22, here we show how a homologous tail fiber from a newly isolated *Alteromonas* schitovirus A5 functions independent of its chaperone. This led us to suggest an alternative function for this chaperone as a transient fiber "cap". In addition, we reveal conservation of synteny, sequence homology, and functional similarity between the host recognition modules (including the tail fibers and chaperones) of A5, V22, and other distantly related phage genomes infecting different *Alteromonas* spp. hosts, which supports transfer of this specific module among these divergent viral genomes. This provides an important example of phage genomic mosaicism and adaptation of host infectivity by importing host recognition modules of other phages already infecting a certain host; in this case the widespread marine bacterium *Alteromonas*.

## Results

### Isolation and characterization of *Alteromonas* phage A5

Bacteriophage vB_AmeP_A5 (phage A5) was isolated from coastal waters collected in Alicante, Spain using *A. mediterranea* strain PT15 (GenBank NZ_CP041170). Plaque formation was not observed against any other *A. mediterranea* or *A. macleodii* strains (Table 1). Phage A5 has a double-stranded (ds)DNA genome of 75,104 bp with 91 predicted coding DNA sequences, no tRNAs, and an overall G+C content of 38.1% (Fig. 1a). In addition, no lysogeny-related genes (coding for, e.g., integrases) could be identified, suggesting A5 is a strictly lytic phage. Sequence analysis identified phage A5 as an Enquatrovirus within the *Schitoviridae* family (previously members of the *Podoviridae* family)[25] with 43.62% sequence similarity to Escherichia phage N4 (28% query coverage), the best-known member of this family (GenBank NC008720). The closest *Alteromonas* phage to A5 was identified as phage vB_AmeP_V19 (54.7% sequence similarity with 15% query coverage) (GenBank OP751378). Phage A5 has podoviral morphology, typified by a short non-contractile tail as observed for other *Schitoviridae* members (Fig. 2c). An interesting feature of these phages is the presence of three predicted RNA polymerase genes (identified in A5 as gp29, gp61, and gp66), including a large (A5 gp29; 3495 amino acids (aa)) virion-encapsidated DNA-dependent RNA polymerase (vRNAP)[26] that is injected into host bacteria during infection along with the phage DNA[25,27]. The alignment of the phage A5 genome to *Escherichia* phage N4 and other *Alteromonas* schitoviruses such as V19 (OP751378), P1 (NC021532)[28] (Fig. 1b) or other schitoviruses (Fig. S1) revealed synteny and sequence similarity across their genomes, e.g., the large terminase (TerL) and the portal and major capsid proteins within the structural cassette. However, no similarity could be identified within the host recognition modules of these phages suggesting the phages analyzed use different recognition and infection machineries.

### A homologous host recognition module among distant *Alteromonas* phage families

To identify potential similarities between the host recognition modules of A5 and other *Alteromonas* phages, genomic alignments were performed between A5 and all *Alteromonas* phages sequenced to date including *A. mediterranea* phage V22[21]. A clear syntenic pattern was found along ten genes located in the structural region of the A5 and V22 genomes, with several protein sequences ranging from 45% to 96% similarity (Fig. 2a). This included the previously characterized tail fiber of V22 (gp26) which shares 35% sequence similarity (60% within its C-terminus, residues 337–464) to its A5 counterpart (gp8) and 56% sequence similarity between their cognate chaperones (V22 gp27 and A5 gp9). Homologs to other tail-related genes of V22 involved in host recognition (e.g., gp24 and gp25) were also identified in phage A5 (gp6 and gp7) (Fig. 2a and Table 2). Downstream, gp11 of A5 shares 66% similarity with gp30 of V22, which was previously identified as a member of the CheY-like family of proteins (IPR001789)[21] involved in signal transduction[29] and suggested to function as a class II auxiliary metabolic gene (AMG), which are used to enhance host functionality during viral propagation. Finally, gp29, gp31, and gp32 of V22 present between 41% to 64% sequence similarity with A5 gp10, gp12, and gp13; however, their functions remain unknown. Further analysis using the non-redundant (nr) NCBI database identified a similar gene configuration and sequence homology between the host recognition modules of A5 and *A. macleodii* phage P24, isolated from coastal waters of the Yellow Sea (China)[30] (Fig. 2a). A similar host recognition module was also identified within a viral metagenome-assembled genome (MAG), GOV_bin_2917[31] (GenBank MK892806), sampled from the Indian Ocean. This module is most similar to that of phage V22;[21]

**Table 1 | Host range analysis of *Alteromonas* schitovirus A5**

| Species | Strain | Infection | GenBank Accession No. |
|---|---|---|---|
| *Alteromonas mediterranea* | PT15 | + | NZCP041170 |
| | PT11 | - | NZCP041169 |
| | CH17 | - | NZCP046670 |
| | DE1 | - | NC019393 |
| | U7 | - | NC021717 |
| *Alteromonas macleodii* | AD45 | - | NC018679 |
| | MIT1002 | - | NZJXRW00000000 |
| | HOT1A3 | - | NZCP012202 |
| | ATCC 27126 | - | CP003841 |
| | BS11 | - | NC018692 |

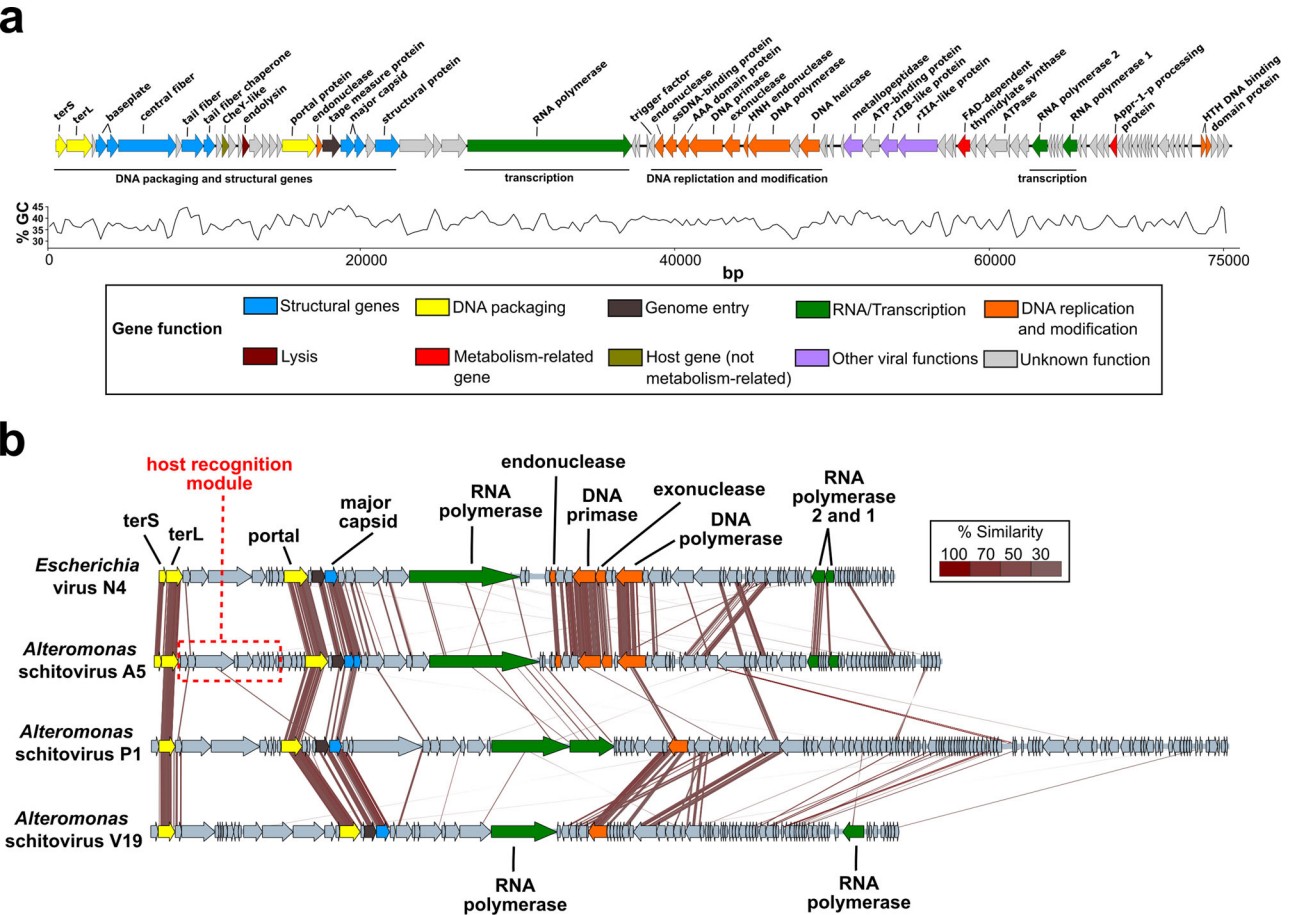

**Fig. 1 | Genomic analysis of phage A5. a** Annotated genome of phage A5 colored based on detected molecular functions of individual gene products. Percentage GC content across the whole genome is displayed. **b** Genome alignments of phage A5 to *Escherichia* phage N4 (NC008720) and *Alteromonas* schitoviruses P1 (NC021532) and V19 (OP751378). Sequence alignment was performed using tBLASTx with 30% minimal similarity on 10 bp minimum alignments.

however, as a MAG, the phage has not been physically isolated or characterized.

Phages A5, V22, and P24 belong to distantly related phage families each featuring a morphologically distinct tail structure: A5 is a schitovirus (of podoviral morphology), V22 is a myoalterovirus (of myoviral morphology), and P24 remains an unclassified *Caudoviricetes* (of siphoviral morphology) (Fig. 2c). This also explains why no other regions of sequence similarity or synteny could be identified within their genomes. A phylogenetic analysis using the terminase (TerL) of all *Alteromonas* phages (Fig. 2b) also showed that the three phages group into independent clades and cluster with members of their corresponding viral families. The three phages were not able to infect the other *Alteromonas* strains tested, with phages A5 and V22[21] shown to only infect *A. mediterranea* strains PT15 and PT11, respectively, and phage P24 reported as only infecting *A. macleodii* strain ATCC 27126[30].

AlphaFold 2.0 and AlphaFold-Multimer[32,33] were used to complement sequence-based determination of functions of certain components of the host recognition modules of A5, V22, and P24 (Figure S2). For instance, the central fibers V22 gp24, A5 gp6, and P24 gp43 were all predicted to form distal homotrimeric β-helical fibers with lectin-like head domains that are typically used by other phage RBPs to recognize saccharidic components of bacterial cell walls[5]. V22 gp23 was modeled with high confidence as a short-tail fiber (STF) with a distal tip analogous to that of the long-tail fiber of phage T4[34], including three HxH metal-binding sites that would typically coordinate $Fe^{2+}$ ions within the center of the fiber. P24 gp41 was also validated as a distal tail protein

(Dit) as it was predicted to form a hexameric ring akin to other known Dit complexes that forms the central "hub" of siphoviral baseplates[3,35] (Fig. S2B).

Overall, phages A5, V22, and P24 are morphologically distinct viruses infecting different species and strains of *Alteromonas* yet appear to maintain a conserved host recognition module. This led us to further investigate the structure and function of the principal components shared across these modules—the tail fibers and chaperones—as these are the primary determinants of phage host range and infection.

## Identification of the chaperone-independent tail fiber of phage A5

Fluorescence binding assays were used previously to demonstrate that GFP-tagged V22 tail fiber (gp26) requires co-expression of a downstream chaperone (gp27) to function and bind to *Alteromonas* cells[21]. Here, similar fluorescence binding assays were used to assess the chaperone dependency of the A5 tail fiber after Ni-NTA purification. Surprisingly, similar levels of binding were achieved for GFP-tagged tail fiber (gp8) produced with or without co-expression of its downstream chaperone (gp9) (Fig. 3a). No binding was observed for the A5 tail fiber (GFP-gp8_gp9 or GFP-gp8) when tested against non-host *Alteromonas* strains or the V22 host *A. mediterranea* PT11, which ruled out the possibility of non-specific interactions by GFP-gp8 in either preparation. The chaperone dependency of the V22 tail fiber was also reconfirmed as the tail fiber only bound *A. mediterranea* PT11 host cells after co-expression with gp27 (Fig. 3a).

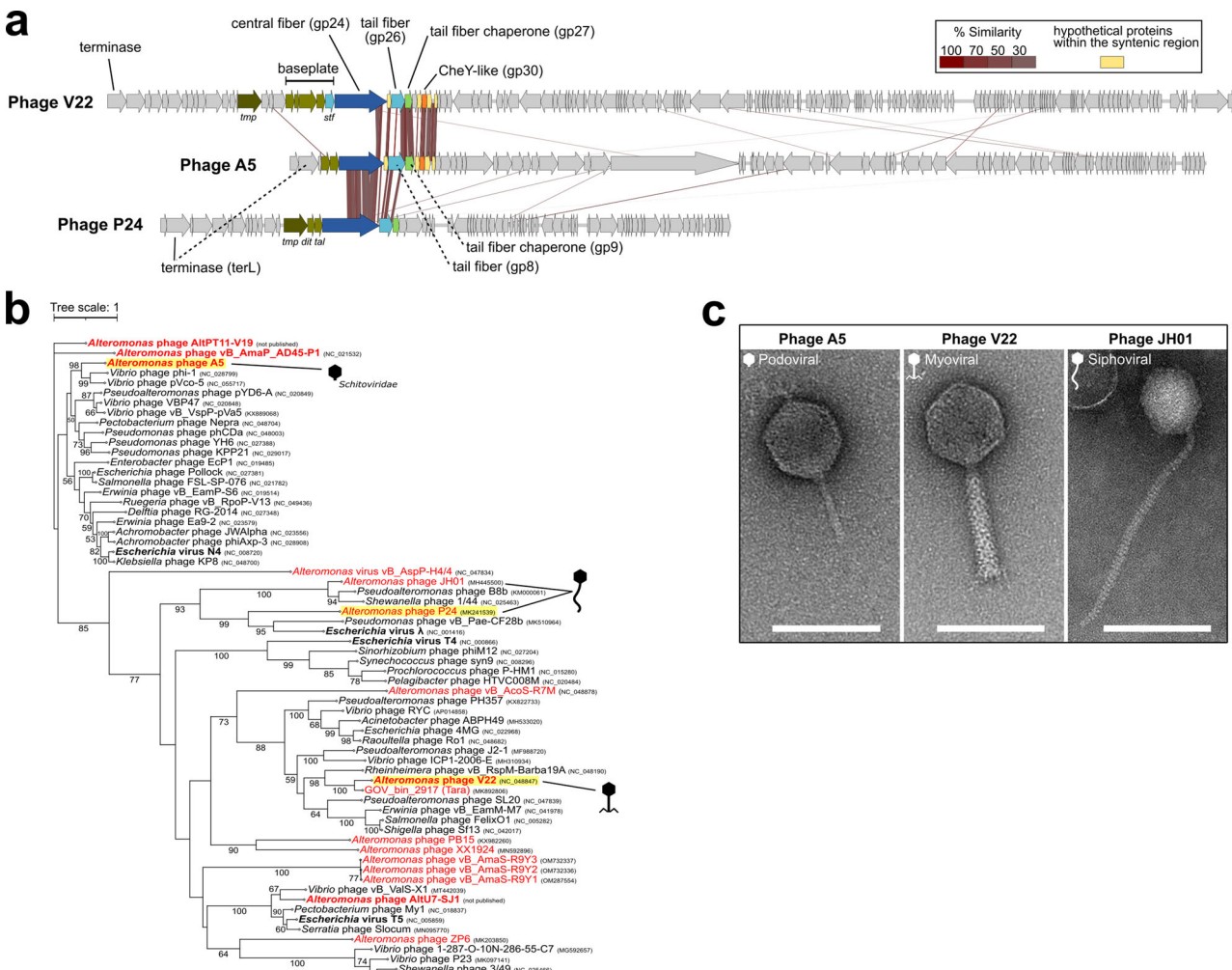

**Fig. 2 | Genomic comparison and terminase phylogeny of phages A5, V22, and P24. a** Genomic alignment of phages A5, V22, and P24 demonstrates synteny and sequence similarity across their host recognition modules. Analysis was performed using tBLASTx with 20% minimal similarity and 10 bp minimum alignment. *tmp*, tape measure protein; *stf*, short tail fiber; *dit*, distal tail protein; *tal*, tail-associated lysin. **b** Terminase (TerL)-based phylogenetic tree groups phages A5, V22, and P24 (highlighted in yellow) into three independent clades together with phages of their own phage families. All *Alteromonas* phages sequenced to date are included in this analysis (highlighted in red). The tree was generated using maximum likelihood and 1,000 bootstrap replications. **c** Transmission electron microscopy (TEM) of phage A5 reveals a podoviral morphology typical for schitoviruses. As expected, TEM of *Alteromonas* phages JH01 (image reproduced from Wang et al.[80]) and V22 reveal a siphoviral and myoviral morphology, respectively. Scale bar (white line), 100 nm. Source data are provided as a Source Data file.

## Variation in chaperone-fiber interactions of the V22 and A5 tail fibers

Co-elution of both A5 gp9 and V22 gp27 chaperones was observed during nickel affinity purification of their respective His-tagged tail fibers (Fig. 3b–d). Previously, the interaction between gp26 and gp27 was shown to be weak as both proteins separated easily during size exclusion chromatography (SEC)[21], which was reconfirmed here (Fig. 3b). In contrast, when chaperone co-expressed A5 tail fiber was analyzed directly after nickel purification the majority of the A5 chaperone remained associated to GFP-gp8 as both proteins eluted as a single peak (Fig. 3c, d; orange line). This was followed by smaller individual peaks of GFP-gp8 (peak 3) and gp9 (peak 4) alone. Complexes formed between chaperones and client proteins (i.e., tail fibers) are multifaceted and can involve different molecular forces (e.g., electrostatic and hydrophobic interactions) that can be influenced by ionic strength[36,37]. Following this reasoning, we used buffer ionic strength to probe the interaction between gp9 and GFP-gp8. The fiber-chaperone complex was incubated in high or low salt (25 mM Tris, pH 7.4, ±500 mM NaCl) after nickel purification for 120 h and analyzed using SEC (Fig. 3c, d). After low salt incubation (green line), GFP-gp8

and gp9 eluted in separate peaks, producing a similar peak profile as observed for V22 tail fiber and gp27. Conversely, after incubation in high salt (magenta line), complex formation between A5 tail fiber and gp9 (peak 2) has been maintained. However, when compared to SEC analysis directly after nickel purification (orange line), the relative amount of tail fiber-chaperone complex had reduced with individual GFP-gp8 and gp9 peaks increasing in size, suggesting slow dissociation of the complex over time. Overall, the interaction between the A5 tail fiber and chaperone appears to be more stable than the transient interaction observed between V22 gp26 and gp27, and it could be partially maintained when kept in a high ionic strength environment, which is closer to the natural (marine) environment where these phages are found.

## Structural analysis of the A5-like extended and V22-like truncated tail fibers

Both the A5 tail fiber and chaperone (gp8, 462 aa; gp9, 240 aa) are larger (by ~25%) than their V22 counterparts (gp26, 369 aa; gp27, 195 aa), which may explain the differences observed in chaperone dependency and tail fiber-chaperone interactions for these otherwise

**Table 2 | Percent similarity in amino acid sequence alignments of host recognition module proteins of phages V22, A5, and P24**

| | | Function | V22 | | | | | | | | | | | A5 | | | | |
|---|---|---|---|---|---|---|---|---|---|---|---|---|---|---|---|---|---|---|
| | | | gp21 Baseplate wedge protein Vibrio phage XM1 gp16 (PDB: 7KH1) (481; 1-480) | gp22 Baseplate protein DUF2612 IPRO21283 (227; 15-72) | gp23 Short tail fiber-AlphaFold-predicted (237; 76-135) | gp24 Central fiber-AlphaFold-predicted (1,430; 457-924) | gp25 Hypothetical protein (107) | gp26 Tail fiber (369; 6-106) | gp27 Tail fiber chaperone DUF4376 IPRO25484(195; 92-184) | gp29 Hypothetical protein (125) | gp30 CheY-like protein IPRO07789(149; 1-120) | gp31 Hypothetical protein (136) | gp32 Hypothetical protein (87) | gp4 Putative Baseplate protein (235) | gp5 Baseplate adapter protein E. coli phage SU10 gp11 (PDB 7Z47) (244; 11-228) | gp6 Central fiber-AlphaFold-predicted(1,234; 540-698, 600-747) | gp8 Tail fiber(462; 57-261) | gp9 Tail fiber chaperone DUF4376IP-RO25484(239; 138-231) |
| A5 | gp4 | Putative Baseplate protein(235) | --- | n.s.s. | --- | --- | --- | --- | --- | --- | --- | --- | --- | | | | | |
| | gp5 | Baseplate adapter protein E. coli phage SU10 gp11 (PDB 7Z47) (244; 11-228) | --- | --- | n.s.s. | --- | --- | --- | --- | --- | --- | --- | --- | | | | | |
| | gp6 | Central fiberAlpha-Fold-predicted(1,234; 540-698, 600-747) | --- | --- | --- | 52% (1,227-1,442) | --- | --- | --- | --- | --- | --- | --- | | | | | |
| | gp7 | Hypothetical protein(119) | --- | --- | --- | --- | 67% (33-114) | --- | --- | --- | --- | --- | --- | | | | | |
| | gp8 | Tail fiber(462; 57-261) | --- | --- | --- | --- | --- | 60% (337-464) | --- | --- | --- | --- | --- | | | | | |
| | gp9 | Tail fiber chaperone DUF-F4376 IPRO25484 (239; 138-231) | --- | --- | --- | --- | --- | --- | 56% (1-243) | --- | --- | --- | --- | | | | | |
| | gp10 | Hypothetical protein(122) | --- | --- | --- | --- | --- | --- | --- | 41% (1-122) | --- | --- | --- | | | | | |
| | gp11 | CheY-like protein IPRO07789(147; 1-119) | --- | --- | --- | --- | --- | --- | --- | --- | 66% (1-149) | --- | --- | | | | | |
| | gp12 | Hypothetical protein(158) | --- | --- | --- | --- | --- | --- | --- | --- | --- | 55% (1-135) | --- | | | | | |
| | gp13 | Hypothetical protein(92) | --- | --- | --- | --- | --- | --- | --- | --- | --- | --- | 64% (10-87) | | | | | |
| P24 | gp40 | Putative tape measure protein(645) | n.s.s. | --- | --- | --- | --- | --- | --- | --- | --- | --- | --- | --- | --- | --- | --- | --- |
| | gp41 | Distal tail protein (Dit) AlphaFold-predicted(195; 1-195) | --- | n.s.s. | --- | --- | --- | --- | --- | --- | --- | --- | --- | n.s.s. | --- | --- | --- | --- |
| | gp42 | Putative tail-associated lysin (Tal) Uni-Ref100_A0A5I6L3-T2(208; 2-187) | --- | --- | n.s.s. | --- | --- | --- | --- | --- | --- | --- | --- | --- | n.s.s | --- | --- | --- |
| | gp43 | Central fiberAlpha-Fold-predicted (1,560; 3-1131) | --- | --- | --- | 45% (1204-1430) | --- | --- | --- | --- | --- | --- | --- | --- | --- | 56% (666-1,486) | --- | --- |
| | gp44 | Tail fiber(361; 10-361) | --- | --- | --- | --- | --- | 43% (8-369) | --- | --- | --- | --- | --- | --- | --- | --- | 51% (1-250) | --- |
| | gp45 | Tail fiber chaperone(176) | --- | --- | --- | --- | --- | --- | 44%(2-156) | --- | --- | --- | --- | --- | --- | --- | --- | 52%(4-49) |

Alignments performed using the BLASTp suite. Highlighted are the putative functions and respective domain IDs as identified by InterPro (when available), HHPred analysis (> 90% probability; PDB IDs provided), and in-house AlphaFold modeling analysis. Brackets provide overall protein length and AA residues for each alignment. *n.s.s.* no significant similarity found. Complementary sequence alignment using tBLASTx is provided in Table S3. Source data are provided as a Source Data File.

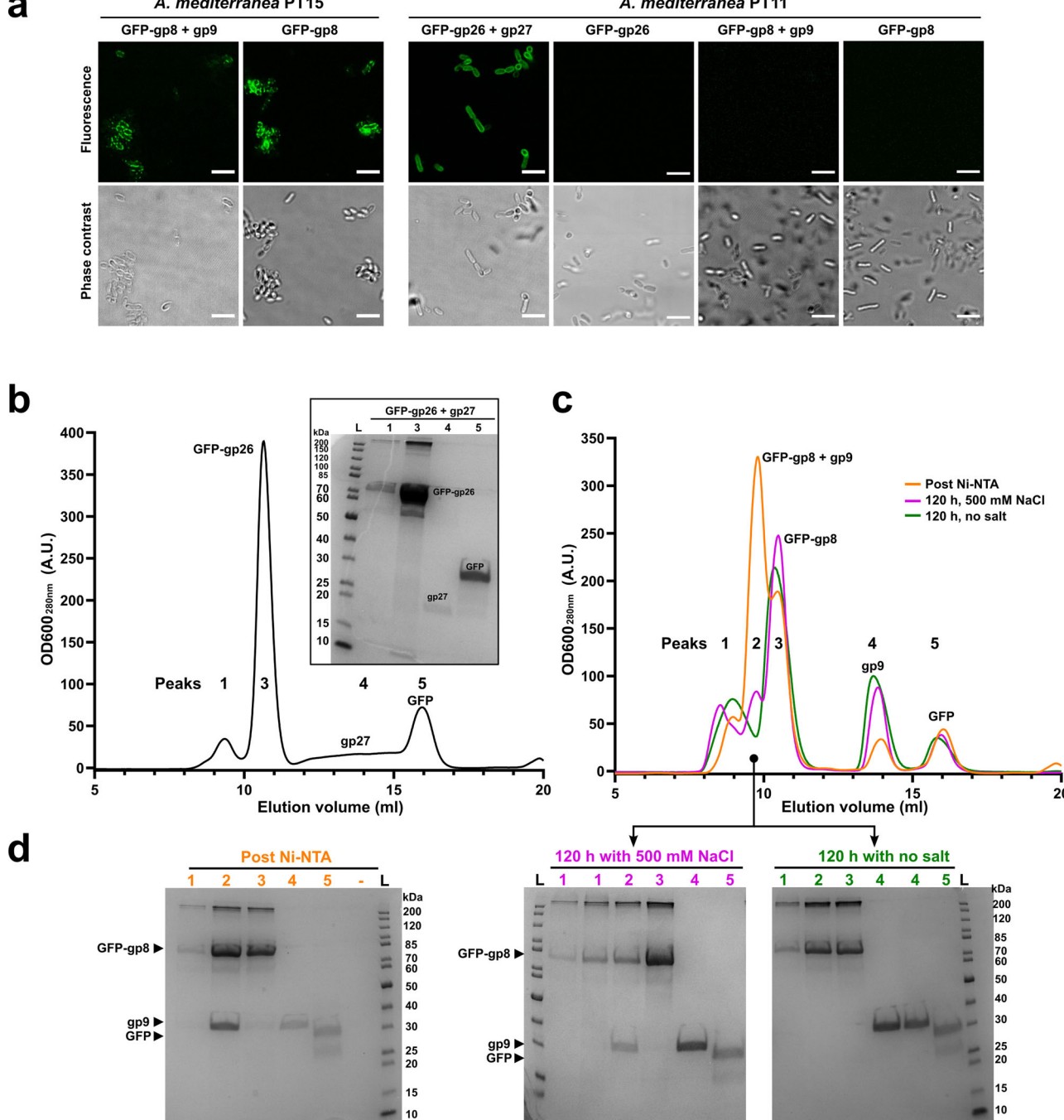

**Fig. 3 | Characterization of the chaperone independent A5 tail fiber. a** Confocal fluorescence microscopy using Ni-NTA purified proteins demonstrating GFP-gp8 tail fiber binds to the phage A5 host *A. mediterranea* PT15 with or without gp9 co-expression, whereas co-expression of gp27 was required to produce GFP-gp26 capable of interacting with the V22 host, *A. mediterranea* PT11. Scale bar, 10 μm. Microscopy experiments were repeated independently at least two times. **b** Size-exclusion chromatography (SEC) elution profile (UV trace at 280 nm) and SDS-PAGE analysis of individual peaks produced by fresh, Ni-NTA purified GFP-gp26 with gp27 co-expression. The main peak 3 contains GFP-gp26 alone, peak 4

contains gp27 alone, and peak 5 contains a GFP side product. **c** SEC elution profiles (280 nm) and **d** SDS-PAGE analyses of GFP-gp8 with gp9 co-expression ran directly after Ni-NTA purification (orange), and after 120 h storage of Ni-NTA eluate in a high (magenta) or low (green) salt environment. For all profiles, peak 2 contained co-eluted GFP-gp8 and gp9, peak 3 contained GFP-gp8 alone, peak 4 contained gp9 alone, and peak 5 contained the GFP side product. Peak 1 in **b** and **c** is void volume. SEC was performed once under specified conditions. Source data are provided as a Source Data file.

highly similar fibers. Sequence alignments including the homologous tail fibers and chaperones of *Alteromonas* phage P24 and MAG GOV_bin_2917 identified a single insertion site flanked by two conserved regions at the center of the A5 and GOV_bin_2917 tail fibers and chaperones (Fig. 4a, b). To investigate the structural significance of this insertion event, AlphaFold 2.0 and AlphaFold-Multimer[32,33] were used

to generate models of the tail fibers and chaperones from "extended" (A5 and GOV_bin_2917) and "truncated" (V22 and P24) variant phages (Fig. 4c, d). All models presented high internal confidence scores with per-residue confidence scores (pLDDTs) between 75.1 and 95.3 and intrinsic model confidence scores (i.e., interface pTM) between 0.46 and 0.67 for the homotrimeric tail fibers, indicating that all proteins

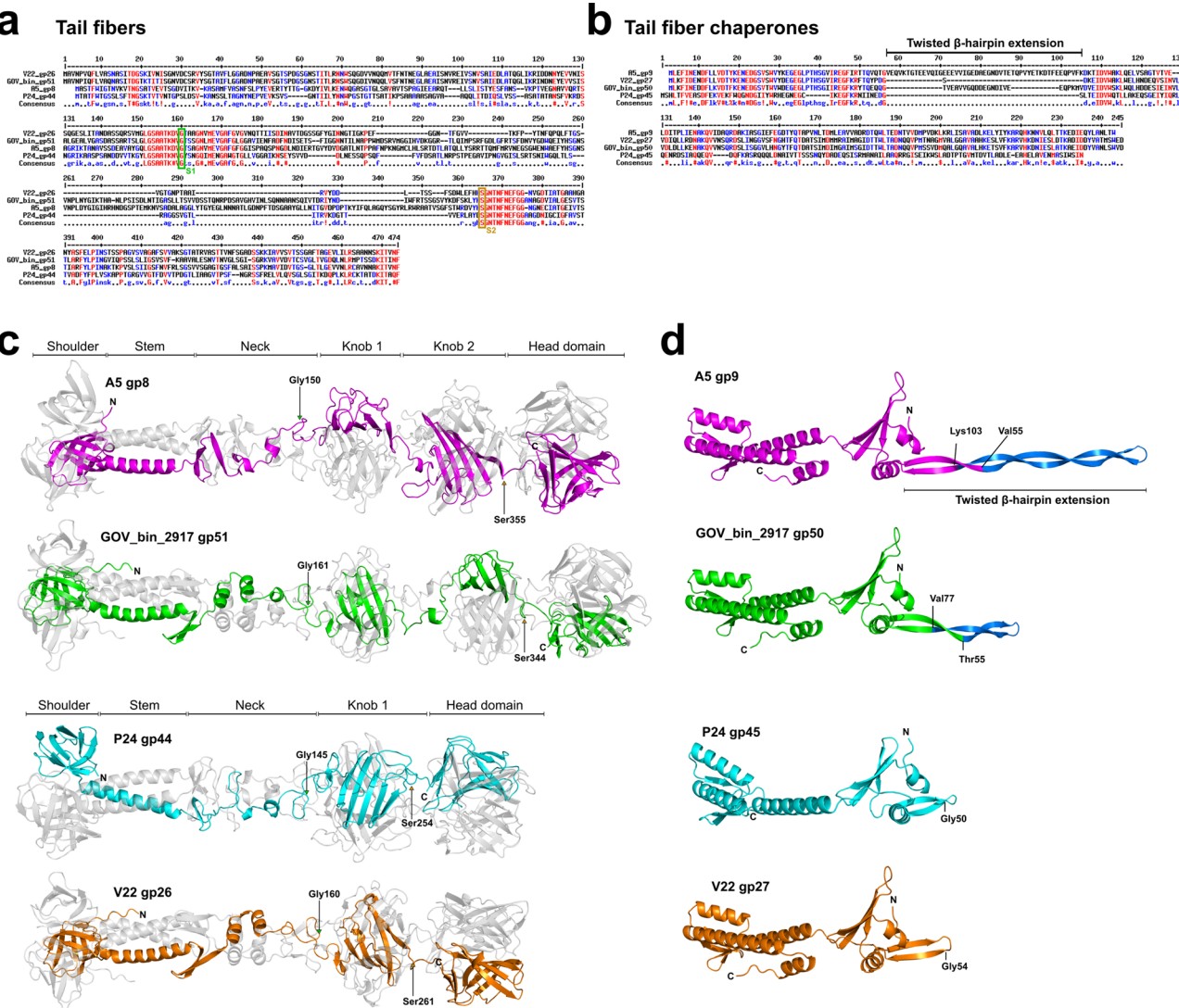

**Fig. 4 | Sequence and structure analysis of extended and truncated *Alteromonas* phage tail fibers and their chaperones.** MultAlin-generated[74] sequence alignments of the four tail fibers (**a**) and chaperones (**b**) with the knob domain(s) flanking residues S1 and S2 of the tail fibers and the twisted β-hairpin extension of the chaperones highlighted, respectively. **c** Ribbon diagrams of full-length, homotrimeric tail fibers predicted using AlphaFold-Multimer. The tail fibers present a conserved domain architecture including an N-terminal shoulder domain, followed by the stem, neck, knob, and distal head domains (the latter assumed to

feature the receptor binding sites for host receptor interactions). Arrows indicate residues S1 (green) and S2 (brown) of conserved flanking sequences of the central knob region for all tail fibers. An insertion in this region results in a knob duplication for the A5 (gp8) and GOV_bin_2917 (gp51) tail fibers. **d** Ribbon diagrams of monomeric, full-length chaperones predicted using AlphaFold 2.0. The chaperones have a conserved architecture except for the twisted β-hairpin extensions (blue) for chaperones A5 gp9 and GOV_bin_2917 gp50. Model confidence scores are provided in Table S1. Source Data (.pdb files) are provided as a Source Data file.

were modeled with high accuracy and suitable for making statements about their structural composition.

Based on architectural similarities with other phage RBPs, distinct and common domains could be identified across all four tail fibers: the N-terminal "shoulder," "stem," and subsequent "neck" domains, followed by central "knob" domain(s), and a distal "head" domain predicted to contain the receptor binding sites of the fibers. The sequence insertion within the A5 and GOV-bin_2917 "extended" tail fibers encodes for an additional knob domain at the center of the fiber (Fig. 4c). The six knob domains from all four tail fibers are all structurally similar while sharing 21 ± 3% sequence similarity (Fig. S3), e.g., the knob domain of V22 superimposes with knobs 1 and 2 of A5 with a root mean square deviation (RMSD) of 1.5 Å and 2.3 Å, respectively. Using the DALI server[38] to identify structural homology, we identified high similarity between the four tail fibers and the fibers of *Pseudomonas aeruginosa* R-type pyocins[39] (Fig. 5a). For instance, both R1 and

R2 pyocins contain similar knob domains as featured in the four phage tail fibers, e.g., knob 2 of the R2 pyocin fiber (PDB ID: 6CL6)[39] aligning with knobs 1 and 2 of A5 and the knob of V22 with Z-scores of 9.2, 8.5, and 8.2 and RMSD of 2.4, 2.6, and 2.4 Å, respectively (Fig. 5b). The DALI server also identified structural similarity between the C-terminal head domains of the four tail fibers with the lectin-like head domains of a putative tail fiber from *Acinetobacter baumannii* phage AP22 (PDB ID: 4MTM; A5 gp8 to AP22 gp53 head domain Z-score of 3.6, RMSD of 2.9 Å)[40] (Fig. 5c), which is a receptor-binding fold commonly found at the distal tips of another phage RBPs[39-42]. Interestingly, the head domains of the central fibers of A5 (gp6), V22 (gp24), and P24 (gp43) were also predicted by AlphaFold as forming similar lectin-like binding domains (Fig. S2A). C-terminal lectin-like domains are also used by the pyocin fibers;[38] however, such structural similarity between the head domains of the A5 and V22 tail fibers was not identified by the DALI server. The universal function of lectin folds for interacting with

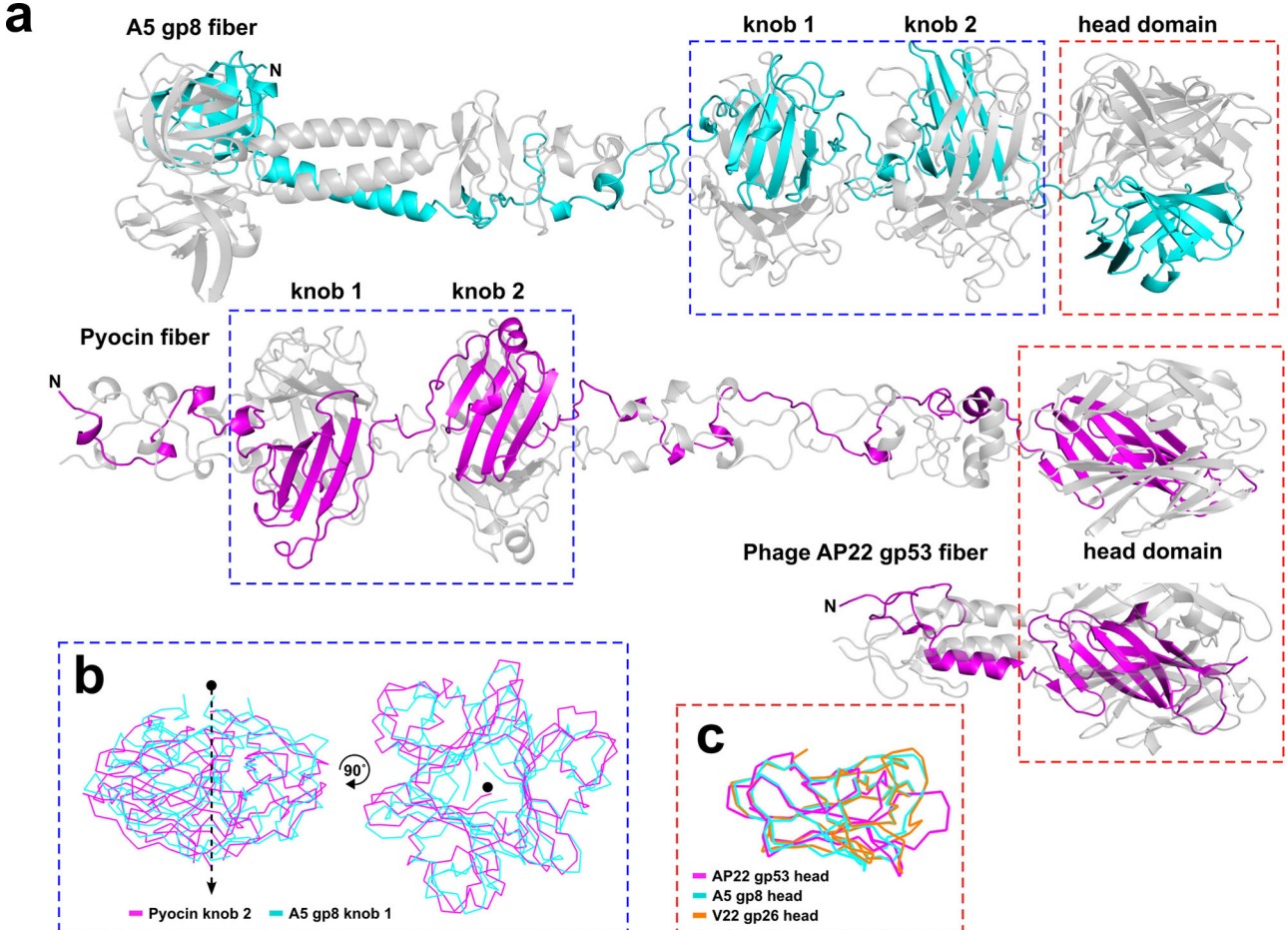

**Fig. 5 | Structural similarities between *Alteromonas* phage tail fibers and other tail fibers. a** Ribbon representations of the A5 gp8 fiber model, the R2 pyocin of *P. aeruginosa* (PDB ID: 6CL6)[39] and the distal tip of the putative tail fiber of *A. baumannii* phage AP22 (PDB ID: 4MTM)[40] contain similar domain structures as highlighted for their knob (blue box) and distal head (red box) domains. **b** Superposition of the A5 knob 1 to the pyocin knob 2 domains as generated by DALI[38] (Z-score=9.2, RMSD of 2.4 Å (78 residues)). Knob domains of V22 and A5 demonstrate similar levels of similarity (Fig. S3). **c** Superposition of the head domains from A5 gp8 and V22 gp26 to that of phage AP22 gp53 (Z-scores of 3.6 and 5.3 RMSD of 2.9 and 2.4 Å) that share the lectin-like fold also featured at the tip of the pyocin fiber. Model confidence scores are provided in Table S1. Source Data (.pdb files) are provided as a Source Data file.

carbohydrate ligands strongly suggests that the *Alteromonas* phage tail fibers and central fibers presented in this work recognize saccharidic components of the *Alteromonas* cell wall; however, the specific receptor and its composition has yet to be realized for these phages.

### A5-like chaperones feature β-hairpin tentacles and closely resemble intramolecular chaperone domains

The insertion within extended chaperones of phages A5 and GOV_bin_2917 introduces an elongated β-hairpin that protrudes from an α-helical body that is shared by all four chaperones (Fig. 4b, d). Overall, the four chaperones share 55.6% sequence similarity and superimpose well (e.g., RMSD of 2.5 Å for A5-gp9 and V22-gp27) with the only non-conformity being the length and composition of the central β-hairpins. All four chaperones closely resemble a class of intramolecular chaperone (IMC) that can be found at the C-terminal ends of RBPs, such as the tail fiber of phage T5 (PDB ID: 4UW8)[43] and the tail spike of phage K1F (PDB ID: 1V0E)[44] (Fig. 6a). The IMCs of K1F TSP and T5 tail fiber form homotrimers via their α-helical cores at the distal tip of their respective RBP with the three β-hairpins (coined "tentacles") extending upwards to embrace the body of the RBP during maturation. A similar IMC complex was predicted to form at the C-terminus of the *Salmonella* phage S16 tail fiber[45], however, with two β-hairpins associated with the fiber body instead of one (Fig. S2E). Normally, after assisting with β-helical formation of the RBP the IMCs are removed by

autoproteolysis. AlphaFold-Multimer predicted a similar homo-trimeric IMC-like complex for A5 gp9 and V22 gp27 (Fig. 6b). In both cases, the α-helical bodies of the three chains formed a central core with their β-hairpins of variable length (Fig. 6c) protruding upwards, mimicking the tentacle domains of the IMCs.

The structural similarity to bona fide IMCs implied that a similar–albeit intermolecular–chaperone-fiber complex may form between the *Alteromonas* phage tail fibers and its chaperones. Such hetero-hexameric complexes were subsequently confirmed for phages A5 and V22 with high confidence using AlphaFold-Multimer (Fig. 6d). In both models, the interaction between individual fiber and chaperone pairs was mainly via hydrogen bonding between β-strand (β4) of the A5/V22 head domains and a β-strand (β3) from their respective chaperone domain, forming a six-strand β-sheet at the center of both complexes (Fig. 6d; insets). Interestingly, the three β-strands of the *Alteromonas* chaperones do not exist in the IMC domains of the T5, K1F, or S16 RBPs, but are conserved in the P24 and GOV_bin_2917 chaperone models, suggesting that the β-sheet region of the *Alteromonas* phage chaperones is co-evolutionary connected to the β-sheet of the fiber head domains.

### Discussion

The marine bacterium *Alteromonas* has a global distribution and is important for carbon and nitrogen cycling in the ocean[46]. Thus,

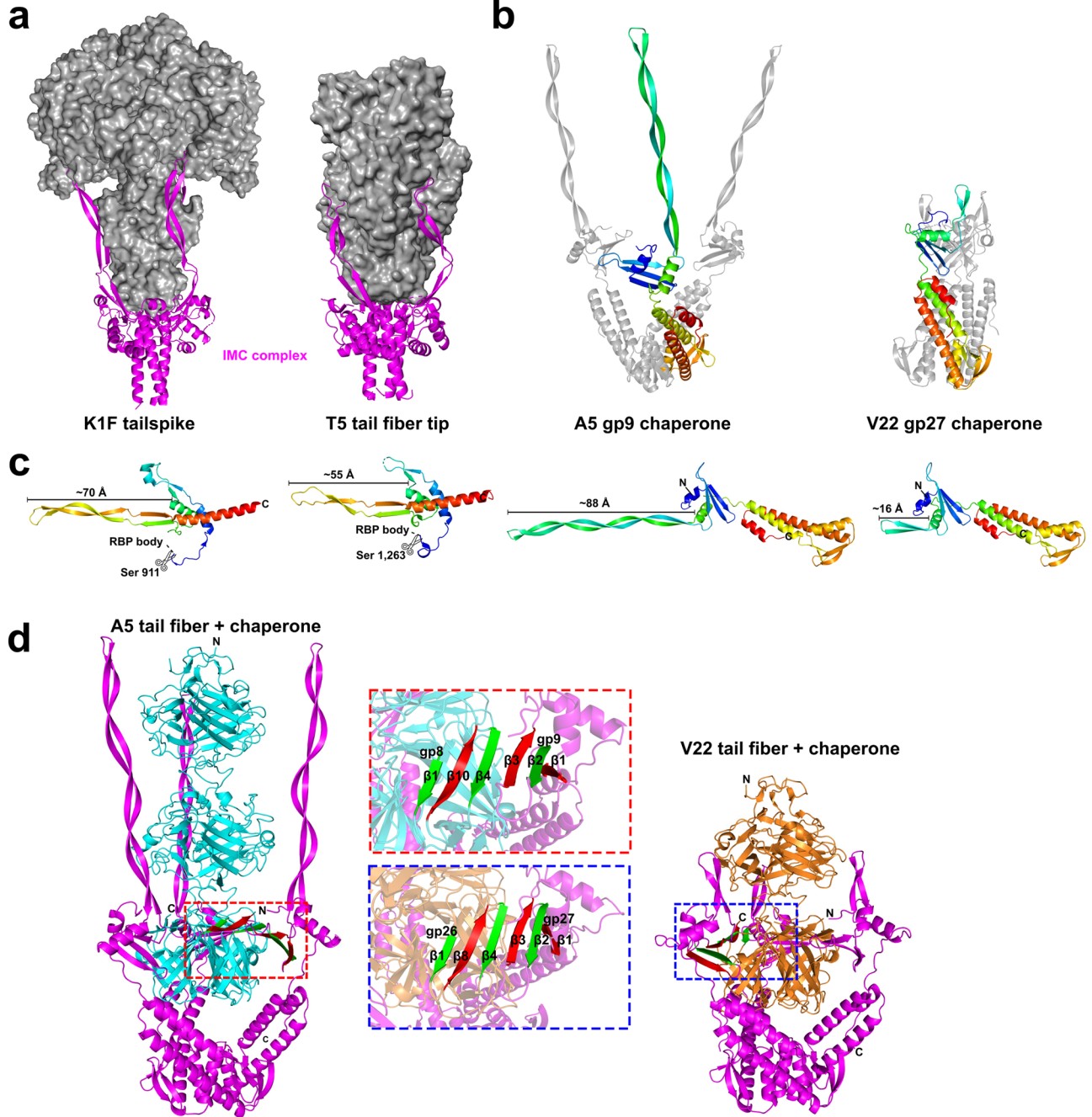

**Fig. 6 | *Alteromonas* phage chaperones resemble intramolecular chaperone domains used by other phage RBPs. a** A5 gp9 and V22 gp27 chaperones closely resemble the C-terminal intramolecular chaperone domains (IMC; magenta) that mediate trimerization and maturation of phage RBPs (grey surface structures shown) prior to autoproteolytic cleavage (after the serine (Ser) residues highlighted in (**c**)) and removal from the mature RBP. Representative IMC-containing RBPs are shown for the tailspike of K1F (modeled here using PDB IDs: 3GW6 and 1V0E)[44] and the tail fiber of *E. coli* phage T5 (PDB ID: 4UW8)[43]. **b** Homotrimers of gp9 and gp27 were predicted by AlphaFold-Multimer to form similar arrangements as the IMC complexes of the K1F TSP and T5 tail fiber. **c** Ribbon representations of A5

gp9, V22 gp27, and isolated IMC domains of the K1F tailspike and T5 tail fiber colored from the N- to C- terminus (blue to red) with approximate lengths of the β-hairpin domains indicated. **d** AlphaFold-Multimer predictions of the hetero-hexameric tail fiber (both truncated to the knob domain) and chaperone (full-length) complexes formed by A5 (gp8 and gp9) and V22 (gp26 and gp27). Highlighted is a six-strand β-sheet generated by tail fiber (β1, β4, β8) and chaperone (β1, β2, β3) β-strands formed between each tail fiber and chaperone pair in both the A5 and V22 complex. Model confidence scores are provided in Table S1. Source Data (pdb files) are provided as a Source Data file.

*Alteromonas* phages are of great significance for the study of marine microbial ecology given the role that phages play in the maintenance of bacterial populations and in the evolution of the host through horizontal gene transfer (HGT)[19,47,48]. Nevertheless, despite an increase in metagenomic studies providing an abundance of genetic and genomic information about marine phages and their bacterial hosts[49],

relatively little is known about their mutualistic interplay and the role that genetic exchange has played in their coevolution.

Here, we describe an interesting case of synteny and conservation of a host recognition module used by *Alteromonas* phages isolated from different marine environments and belonging to highly divergent families. Schitovirus A5, myoalterovirus V22, and unclassified

*Caudoviricetes* (siphoviral morphology) P24 share a common host recognition module with 58.8% sequence similarity with no synteny or sequence homology found across the rest of their genomes (Fig. 2a). Each module features essential components related to their morphologically distinct infectious machineries; however, all receptor binding elements, including the central fibers and bona fide tail fibers (as demonstrated here) are clearly conserved. Phage genomes have a highly mosaic composition[50], with each phage considered a unique combination of interchangeable modules that are transferrable between phages by HGT and recombined through illegitimate and homologous recombination[17]. These processes are essential for maintaining genetic and phenotypic diversity of phage populations as they co-exist alongside their continually adapting bacterial hosts[47,51–54]. There are a few documented cases of homologous RBPs, or shared receptor binding domains used by morphologically distinct phages that provides strong evidence of such gene transfer events across morphotype borders[55–58]. Interestingly, our identification of homologous host recognition modules (including tail fiber and chaperone pairs, central fibers, CheY-like auxiliary metabolic genes, and other components) used by phages A5, V22, and P24 suggests that HGT and recombination events may be a more common process than previously thought across the morphotype classes of the *Caudoviricetes* (tailed) phages.

Supposedly, several transfer events of these host recognition modules took place a long time ago between ancestors of these three phages, generating an evolutionary advantage that was eventually fixed throughout successive clonal lineages. As proposed by Hendrix[15] and others[16,17], the current recognition modules of these phages may be the result of ancient illegitimate (non-homologous) recombination events, followed by extended periods of natural selection and mutation that restructured each module configuration. Another possibility is that these recognition modules might have been obtained from other unrelated phages with similar molecular targets, providing a swift change in the host range and a way to compensate for host evasion by receptor exchange[8].

As the main determinant of host range and phage infectivity, we focused our efforts on deciphering the structure and function of the primary tail fiber and its cognate chaperone that is the central component shared among the host recognition modules of these phages (as well as the MAG GOV_bin_2917[31]). Interestingly, AlphaFold model predictions identified two variants of tail fiber and chaperone pairs: the "extended" A5-like and the "truncated" V22-like (Fig. 4). The A5-like tail fibers featured a central knob domain duplication which correlates with an extension of a central β-hairpin domain of its paired chaperone. The compositional similarity between both chaperone variants and the intra-molecular chaperone (IMC) domains of previously characterized RBPs implied the formation of a similar complex at the tip of the *Alteromonas* phage tail fibers, which could be confirmed using AlphaFold-Multimer[33] as shown in Fig. 6d. The A5 and V22 tentacles also contain a high proportion of polar residues, similar to those found in the tentacles of the K1F and T5 IMCs. This similarity would facilitate a comparable network of hydrophilic interactions with the central RBP, as previously described[43,44]. However, despite the A5 and V22 complexes being predicted with high confidence, the tentacles are modeled as perpendicular extensions to the fiber and are too distant to suggest a network of interactions between the chaperone and fiber tip. Nevertheless, a clear interaction between both fibers and their respective chaperones was identified by the concatenation of a six-strand β-sheet via hydrogen bonding at the center of both complexes. The lack of similar β-strands within the IMCs of other RBPs, strongly suggests that this region has been conserved as a fiber interaction site by such intermolecular chaperones, which likely co-evolved with the corresponding β-strand region of fiber head domains.

The requirement of gp27 co-expression to produce functional V22 tail fibers (gp26), as well as the co-elution of gp27 after nickel affinity purification of His-tagged gp26, provides clear evidence that gp27 is an essential fiber chaperone (Fig. 3b)[21]. On the other hand, despite being homologs, a similar fiber-chaperone relationship was not observed for phage A5. Instead, recombinant A5 tail fiber (gp8) demonstrated similar levels of host cell binding with or without co-expression of its chaperone (gp9). Interestingly, gp9 features the longest β-hairpin (~88 Å) of all the chaperones we modeled in this study (including the T5 and K1F IMC). Based on models of the *Alteromonas* tail fibers and chaperones interacting analogous to the T5 and K1F IMCs, whose β-hairpin "tentacles" lock into the RBP tip, it can be inferred that a longer β-hairpin provides a greater surface area for complex formation. This would explain the relatively higher amount of gp9 that remains associated with the gp8 tail fibers after Nickel purification compared to its V22 counterparts that feature a smaller (~16 Å) β-hairpin (Fig. 3c, d). The incongruity of phage A5 encoding a chaperone with stronger tail fiber binding properties, but is also not required for producing functional tail fibers, suggests that these chaperones may have an alternative function besides assisting with tail fiber maturation. An alternative function has been described for other RBP chaperones, for instance, Tfa$_{Mu}$ remains bound to the tail fiber of phage Mu after assisting with fiber formation and has been suggested to interact with the same lipopolysaccharide receptor as the mature fiber[24].

We propose that the *Alteromonas* phage chaperones could function as transient "caps" to provide temporal shielding of the tail fiber tips. During phage lysis, such caps could block the tail fiber receptor binding sites from interacting with cellular debris or interacting with adjacent bacterial cells, thus modulating fiber interactions during the initial stages of progeny release. Such a capping function is evidenced by the relatively weak chaperone-fiber interaction (for both A5 and V22) combined with the compositional and functional similarity of the chaperones to known IMCs. Owing to the weak interaction between chaperone and tail fiber, a phage particle may lose its chaperones sporadically after progeny release, resulting in a heterogenous population of phages with "capped" and "uncapped" fibers. As demonstrated here, AlphaFold[32] and other artificial intelligence-driven protein modeling tools have greatly improved our ability to interpret protein functions and characterize new phage RBPs and their receptor binding domains based on amino acid sequence alone. Nevertheless, determining how these individual components assemble to form their intricate distal tail machines remains a mystery requiring empirical experimentation. Further investigations are necessary to (i) unravel how the secondary capping function of these chaperones mediates receptor blocking and infectivity of whole phage particles, (ii) identify the capping mechanisms of fiber-chaperone pairs from other phages, and (iii) explain how these homologous host recognition module components can assemble into the three diverse infectious machineries used by these distantly related *Alteromonas* phages.

## Methods

### Bacterial strains, media, and growth conditions
*Alteromonas* spp. strains used in this study are listed in Table 1. *Alteromonas* spp. were grown in marine media (MM) (3.5% sea salts (Sigma), 0.1% yeast extract (Scharlau) and 0.5% peptone (PanReac-AppliChem)) at 25 °C with agitation. *E. coli* XL1-Blue MRF′ cells (Stratagene) were used for all cloning steps, plasmid transformations, and protein expression, and were grown using LB media incubated at 37 °C under agitation.

### Phage isolation
*Alteromonas* phage vB_AmeP_A5 (A5) was isolated from filtered (0.22 μm) Mediterranean coastal waters collected during the summer of 2016 in Alicante, Spain. In brief, the filtered seawater was mixed with MM, spiked with *Alteromonas mediterranea* PT15 (GenBank NZ_CP041170; isolated during the same sampling campaign), and incubated overnight at room temperature. The mixture was re-filtered, and

the same procedure repeated two more times. After the third round of phage propagation, individual plaques were collected, passaged on strain PT15 three times using the double agar overlay technique, and finally propagated on a large scale[59] using a multiplicity of infection (MOI) of 0.1. Phage stocks were filtered (0.22 μm), dialyzed into SM buffer (50 mM Tris, 100 mM NaCl, 10 mM magnesium sulfate, pH 7.5), and titered (plaque forming units (PFU)/mL) using the double agar overlay technique. Long-term stocks were established at 4 °C and −80 °C (with 20% glycerol added).

## Phage A5 gDNA extraction

For gDNA extraction, 12.5 mM of $MgCl_2$, 5 U of DNase I (Thermo Scientific), and 0.3 mg/ml RNase A (Thermo Scientific) were added to 1 mL of phage stock ($2.3 \times 10^{10}$ PFU/mL) and incubated for 30 min at room temperature. 20 mM EDTA, 0.05 mg/ml proteinase K (PanReac-AppliChem) and 0.5% SDS were added and incubated at 55 °C for 1 h. Phage DNA extraction was performed using the phenol-chloroform method and precipitated with 0.1 volumes of 3 M sodium acetate and 2.5 volumes of 100% ethanol. Purified gDNA was resuspended in 50 μL Milli-Q water, quantified using a Qubit® 3.0 fluorometer (Invitrogen), and stored at −20 °C.

## Sequencing, annotation, and genome visualization

Phage A5 gDNA was sequenced using an Illumina MiSeq instrument (2 × 300 bp) at the FISABIO facilities (Valencia, Spain). Sequenced reads were quality-checked using FastQC v0.11.7[60] and quality trimmed using Trimmomatic v0.32[61]. Assembly of the reads was performed using SPAdes v3.12.0[62]. Coverage analysis of the reads was analyzed with Bowtie2 v1.3.1[63], SAMtools v1.2[64], and Tablet[65] using the obtained contigs as a reference. Prodigal[66] was used to predict open reading frames (ORFs), and gene annotation was carried out using Diamond v0.9.4.105[67] against the non-redundant (nr) NCBI database (e-value < 1e-3) and hmmscan (HMMER v3.1b2)[68] against pVOGs and Pfam databases[69,70] (e-value < 1e-3). Individual CDSs of interest were manually inspected using the Conserved Domain Database suite (CDD/SPARCLE)[71], InterPro[72], and the HHpred server[73]. Easyfig v2.2.2[74] and in-house python software were used for genome visualization and genome comparison. Sequence alignments were performed using MultAlin[75].

## Phylogenetic analysis

Amino acid sequences of the large terminase (TerL) from all *Alteromonas* genomes available to date and similar TerL sequences infecting other bacteria (53 sequences in total) as well as TerL sequences from phages T4, λ, T5, and N4 (outgroup) were extracted from the non-redundant NCBI database. Sequences were aligned using MUSCLE[76] and a phylogenetic tree was generated using IQ-TREE v1.6.11[77]. ModelFinder[78] identified VT, F, and R4 models as the most appropriate for phylogeny reconstruction, and a tree was generated using maximum likelihood and 1000 bootstrap replications. The final tree was visualized using iTOL v4[79].

## Genome comparative analysis

All *Alteromonas* viruses present in the NCBI database were used for comparative analysis with *Alteromonas* phages A5 and V22. All *Alteromonas* schitoviruses were selected for comparison to the phage A5 genome using the *Escherichia* virus N4 as a model (representative alignments shown in Fig. S1). Genomic alignments presented in Figs. 1b and 2a were performed using in-house python software with tBLASTx and 30% minimal similarity on 10 bp minimum alignments. For Fig. S1, in-house Perl software was used selecting tBLASTx and 30% minimal similarity on 100 bp minimum alignments. Synteny was considered when the genomic region showed at least five or more consecutive syntenic genes (i.e., similar size and gene order) with a maximum separation of four non-syntenic genes.

## Structure prediction and analysis

Structure predictions were performed with AlphaFold 2.0[32] and AlphaFold-Multimer[33] downloaded from www.github.com/deepmind/alphafold and installed on a HP Z6 workstation equipped with a Xeon Gold 6354 CPU, 192 GB of RAM, an Nvidia RTX 2080TI GPU, and M2 SSD disks, running Ubuntu Linux 20.04. All predictions were assessed using internally generated confidence scores. Confidence per residue is provided as a predicted Local Distance Difference Test score (pLDDT; scored 0–100), with the average of all residues per model presented in Table S1. A pLDDT ≥90 have very high model confidence, residues with 90> pLDDT ≥70 are classified as confident, while residues with 70>pLDDT >50 have low confidence. Interface pTM scores (iptm+ptm) are a measure of predicted structure accuracy generated by AlphaFold-Multimer and provide the overall confidence score for the complete model (scored 0 to 1). All structure figures presented in this work were produced using Pymol (PyMOL Molecular Graphics System, version 2.4.1, Schrodinger LLC). DALI server was used to identify structural homologs in the PDB[38].

## Phage host range

The host range of phage A5 was tested against ten *Alteromonas* spp. strains (Table 1) using a spot test infection assay. In short, 10 μL of a phage A5 dilution series in SM buffer ($10^{10}$ to $10^5$ PFU/ml) or 10 μL of SM buffer alone (control) were spotted on a bacterial lawn containing 100 μL of an overnight bacterial culture mixed with 3 mL of 0.7% marine agar and incubated for 16 h at 25 °C. Formation of individual plaques confirmed the successful infection of the host.

## Transmission electron microscopy

Phage particles were concentrated and purified using CsCl isopycnic centrifugation and dialyzed into SM buffer to reach ~$10^{11}$ PFU/ml. Phage particles were negatively stained for 20 s with 2% uranyl acetate on carbon-coated copper grids (Quantifoil) and observed at 100 kV on a Hitachi HT 7700 equipped with an AMT XR81B Peltier cooled CCD camera (8 M pixel) at the ScopeM facility, ETH Zurich, Switzerland.

## GFP-tagged tail fiber plasmid construction, expression, and purification

Selected gene fragments (*gp8* and *gp8_gp9*) were inserted into a pQE30 derivative plasmid pQE30_HGT (featuring an N-terminal His-tag connected to GFP via a TEV-cleavage site[41]) using the Gibson isothermal method (NEBuilder HiFi DNA assembly master mix, New England Biolabs). Gene fragments were generated by PCR using primers (Table S2) and A5 phage particles or pQE30_HGT as DNA templates. Assembled plasmids were transformed into *E. coli* XL1 Blue MRF′ cells, purified, and verified by Sanger sequencing. Individual plasmids were transformed into *E. coli* XL1 Blue MRF′ cells and grown in LB media supplemented with 100 μg/ml ampicillin and 15 μg/ml tetracycline with agitation at 37 °C until reaching an $OD_{600}$ of 0.6. Cultures were cooled to 20 °C, induced with 0.5 mM isopropyl-β-$_D$-thiogalactopyranoside (IPTG), and incubated for 16 h with agitation at 20 °C. Cells were harvested by centrifugation at 5,500 x *g* for 15 min, resuspended in 30 ml of Buffer A (50 mM $Na_2HPO_4$, pH 8.0, 500 mM NaCl, 5 mM Imidazole, 0.1% Tween-20) at 4 °C, and lysed using a Stansted pressure cell homogenizer (Stansted Fluid Power). Cell extracts were centrifuged to remove cell debris at 15,000 x *g* for 60 min prior to immobilized-metal affinity chromatography (IMAC) using low-density Ni-nitrilotriacetic acid (NTA) resin (Agarose Bead Technologies). Buffer A was used to wash beads before eluting proteins using Buffer A + 250 mM imidazole. Proteins were subsequently dialyzed for 16 h into 25 mM Tris-HCl, pH 7.5 and stored at 4 °C unless otherwise stated in the results.

## Fluorescence microscopy

500 μl of overnight bacterial cultures were pelleted by centrifugation (6000 x *g*, 5 min) and resuspended in fresh MM to an $OD_{600}$ of 0.5.

50 µg of Ni-NTA purified protein was added to the cells and mixed using an overhead rotator for 30 min at room temperature. Cells were collected by centrifugation (6000 g, 5 min), resuspended, and washed in 1 ml of MM, collected again by centrifugation, and resuspended in 200 µL of MM. For fluorescence microscopy, 4 µL of the cell suspension was imaged using a confocal inverted microscope (Leica TCS SPE) equipped with an ACS APO 63x/1.30 OIL CS objective lens with excitation at 488 nm and emissions collected with a PMT detector in the detection range of 510 to 550 nm. Transmitted-light microscopy images were obtained with the differential interference contrast mode. Images were acquired with a Leica DFC 365 FX Digital Camera controlled with the LAS AF software. Fiji v2.0.0 (ImageJ software) was used to produce the final microscopy images. To improve visualization of cells, brightness and contrast of only the phase contrast micrographs were auto-adjusted using ImageJ.

### Recombinant protein size exclusion chromatography

0.5 mg of Ni-NTA protein eluate was analyzed by size exclusion chromatography (SEC) using a Superdex 200 10/300 column (GE Life Sciences) in 25 mM Tris-HCl, 500 mM NaCl, pH 7.4 on an ÄKTA purifier FPLC (Amersham Biosciences) with 0.5 ml/min flow speed. Peaks were detected at wavelengths of 280 nm, 260 nm, and 315 nm and collected separately using a Frac-950 fraction collector (Amersham Biosciences). Protein content of each peak was analyzed by SDS-PAGE. 20 µg of purified protein with Laemmli sample buffer (BioRad) and treated with or without heat denaturation (96 °C, 8 min) and ran on a TGX stain-free precast gel (Bio-Rad) for 38 min at 200 V using PageRuler Unstained Protein Ladder (Thermo Scientific). Protein bands were visualized using both UV absorbance (280 nm) and InstantBlue Coomassie staining (Expedeon) on a Gel Doc XR+ imaging system (Bio-Rad). To investigate chaperone-fiber interactions, Ni-NTA purified GFP-gp8_gp9 was dialyzed into high and low salt buffers (25 mM Tris-HCl, pH 7.4, ±500 mM NaCl) and stored at 4 °C for 120 h prior to a repeat of SEC using 0.5 mg protein and the high salt running buffer.

### Statistics and reproducibility

A single transmission electron microscopy image of phages A5 and V22 (Fig. 2C) was selected from multiple micrographs taken from a single phage-coated grid as is common practice. Phage infectivity assessment was performed at least two times producing the same results. Fluorescence microscopy was independently repeated at least two times per experiment (protein+strain) providing similar results that were equivalent to previous observations[21]. SEC chromatograms were collected once under specified conditions in a single session, with each protein run in consecutive order. Analogous results were observed independently and matched previous observations for V22 gp26-27[21].

### Reporting summary

Further information on research design is available in the Nature Portfolio Reporting Summary linked to this article.

## Data availability

Annotated genomes of vB_AmeP_A5 and its host *A. mediterranea* PT15 are available from the GenBank database under accession numbers OP481051 and NZ_CP041170, respectively. *Alteromonas* phage vB_AmeP_V19 accession number is OP751378. Source data are provided with this paper.

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

## Acknowledgements

We thank Allison Coe and Daniel Sher for providing *Alteromonas* spp.
strains MIT1002 and HOT1A3. We thank Asier Zaragoza-Solas (University
of Vienna) for the development of the Python software used to generate
Figs. 1b and 2a. We thank Petr G. Leiman (The University of Texas Medical
Branch at Galveston) for scientific discussion and for providing access to
their AlphaFold installation to generate models of the A5 (gp8, gp9) and
V22 (gp26, gp27) heterohexamers (Fig. 6). We thank Martin J. Loessner
(ETH Zurich) for access to equipment and resources, Jochen Klumpp
(ETH Zurich) for installation and operation of AlphaFold, and the ScopeM
facility (ETH Zurich) for access to transmission electron microscopy. This
work was supported by grants VIREVO CGL2016-76273-P [MCI/AEI/
FEDER, EU], (co-funded by FEDER) from the Spanish Ministerio de
Ciencia e Innovación and HIDRAS3 PROMETEO/2019/009 from Gen-
eralitat Valenciana. R.G.S was supported by a Predoctoral Fellowship
from the Valencian Consellería de Educació, Investigació, Cultura i
Esport (ACIF/2016/050) and was a beneficiary of the BEFPI 2019 Fel-
lowship for predoctoral stays from Generalitat Valenciana and The Eur-
opean Social Fund. M.D. was supported through a Sinergia grant
(CRSII5_189957) from the Swiss National Science Foundation (SNSF).

## Author contributions

Conceptualization, M.D., and R.G.S; Funding acquisition, F.R.V. and
R.G.S.; Investigation, R.G.S., M.D., A.B.M.C., J.J.R.G. and R.R.; Project
administration, M.D. and F.R.V.; Resources, M.D. and F.R.V; Supervision,
M.D., F.R.V, and R.R.; Validation, M.D. and R.G.S.; Visualization, M.D. and
R.G.S.; Writing—original draft, M.D. and R.G.S.; Writing—review and
editing, M.D., R.G.S., F.R.V., R.R. and A.B.M.C.

## Competing interests

M.D. is an employee of Microeos GmbH developing bacteriophage-based
antimicrobials. The other authors declare no competing interest.

## Additional information

**Supplementary information** The online version contains
supplementary material available at

Matthew Dunne.

**Peer review information** *Nature Communications* thanks Mikael Skurnik,
and the other, anonymous, reviewer(s) for their contribution to the peer
review of this work. A peer review file is available.

