## [Peer Review File · Nature Communications]

Distantly related *Alteromonas* bacteriophages share tail fibers exhibiting properties of transient chaperone “caps”Reviewers' Comments:

Reviewer #1:

Remarks to the Author:

Gonzalez-Serrano et al. describe the conservation of a host recognition module across taxonomic borders and even morphotypes, whereas the same host recognition module is absent in close homologs of the same phages. Horizontal exchange of host recognition modules is common in rapidly evolving phages, but the authors also observed remarkable structural differences in tail fibers and their cognate chaperones for phage A5 and V22 (and their homologs). These structural differences are linked to differences in the observed interactions between the respective tail fibers and their cognate chaperones. I share the excitement of these observations that hint on a mechanistically different organization and role, however, I feel the manuscript remains too speculative and the authors were not yet able to sufficiently support the suggestions with wet lab experimental data (in spite of the vast data set of predicted structures of high quality). It remains therefore a bit too much 'may', 'suggests', ... The strain panel used to assess the specificity is rather limited to take general conclusions (particularly to support the claim that the host spectrum is 'extremely narrow' in line 164. In fact, one could also say that 20% of the strains tested can be infected and this would be broader than most phages do). The interaction analysis is based on SEC with/without salt, but more advanced/quantitative interaction assays could be considered to support the claims. It also remains an unsolved question why the chaperone of A5 is not essential in spite of its stronger TF interactions. The suggestion of a transient cap is again highly interesting, but due to the lack of experimental data, it remains speculative. Some similar experiments as with TfaMu could have shed further insight. In retrospect, I feel that the title "... tail fibers with transient chaperone caps" was shaping higher expectations than were delivered.

Minor comments:

Line 123: three predicted RNA polymerase genes (since no experimental evidence is given)

Figure 2: For phage V22 and phage P24, the old taxonomy is used in contrast to phage A5.

Line 130: In fact, this is not surprising but frequently observed before for other phages.

Figure 3: The phase contrast settings do not seem to be consistent across the different panels.

Line 264-265: 'strongly suggests' would be more correct than 'strongly implies'

Reviewer #2:

Remarks to the Author:

The manuscript of Gonzalez-Serrano et al utilizes in a sophisticated manner the modern bioinformatics tools to elucidate the functions of different components of bacteriophage tail fibers. The manuscript is very well written and enjoyable reading. The authors nicely demonstrate that host recognition modules of even distantly related phages are similar, and very likely spread via horizontal gene transfer. I don't identify any (severe) flaws in the manuscript.

I have just a few minor points.

L114. Better: (coding for, e.g., integrases)

L414. dialyzed

Figure 1 legend. ... colored based on detected molecular functions of individual gene products.

Sincerely

Mikael Skurnik

Reviewer #3:

Remarks to the Author:

In this well-written and enjoyable to read manuscript, Gonzalez-Serrano et al. study and compare the genomes of several *Alteromonas* phages and find highly homologous receptor-binding proteins between otherwise genetically and morphologically different bacteriophages.

They analyse the structures of the tail fibres and chaperones as predicted by AlphaFold-Multimer and find a more than passing resemblance with pyocin fibres and other bacteriophage receptor-binding proteins. In particular, a lectin-like fold containing head domain, trimeric knob domains

and an alpha-beta chaperone fold, with in some cases an extended beta-hairpin loop that presumably wraps around the fibre.

This is a well-performed study, with sound methodology and no apparent flaws in data analysis. They draw some very interesting but not completely new conclusions from the work:

- the conservation/horizontal transfer of host cell receptor binding units between very different phages, and for the biologically important *Alteromonas* phage is very important, although similar to other cases, as correctly referenced in the manuscript by the authors.
- the potential function of the chaperone not only in folding, but as a "cap" on the tail fibre to prevent premature binding to cell membranes of the host cell. I wonder if an experiment to further prove this function might be possible.

An interesting point is the mechanism of exchange, as discussed on lines 331-340. Would it be possible (now, or at some point in the future when even more phage and host sequences will be available), to discriminate between the ancient and more recent mechanism? Perhaps via some kind of genetic, sequence or structural signature? Presumably these events are too rare, but if the recent mechanism is true, could it be observed in the lab when infecting bacteria containing prophages with a suboptimal new phage - in a many-generation experiment?

Minor comments:

- I would avoid the abbreviation TF and just write tail fibre in full, some readers might confuse TF with for example transcription factor.
- perhaps include a suggestion that if salt partially prevents dissociation of the RBP-chaperone complex (gp8-gp9), the interactions between them are mainly hydrophobic? This might also be apparent from a structural model of the complex?

A host recognition module shared among distant *Alteromonas* bacteriophage families features tail fibers with transient chaperone “caps”

- Response to Reviewers -

Reviewer #1 (Remarks to the Author):

Gonzalez-Serrano et al. describe the conservation of a host recognition module across taxonomic borders and even morphotypes, whereas the same host recognition module is absent in close homologs of the same phages. Horizontal exchange of host recognition modules is common in rapidly evolving phages, but the authors also observed remarkable structural differences in tail fibers and their cognate chaperones for phage A5 and V22 (and their homologs). These structural differences are linked to differences in the observed interactions between the respective tail fibers and their cognate chaperones. I share the excitement of these observations that hint on a mechanistically different organization and role, however, I feel the manuscript remains too speculative and the authors were not yet able to sufficiently support the suggestions with wet lab experimental data (in spite of the vast data set of predicted structures of high quality). It remains therefore a bit too much ‘may’, ‘suggests’, ... The strain panel used to assess the specificity is rather limited to take general conclusions (particularly to support the claim that the host spectrum is ‘extremely narrow’ in line 164. In fact, one could also say that 20% of the strains tested can be infected and this would be broader than most phages do). The interaction analysis is based on SEC with/without salt, but more advanced/quantitative interaction assays could be considered to support the claims. It also remains an unsolved question why the chaperone of A5 is not essential in spite of its stronger TF interactions. The suggestion of a transient cap is again highly interesting, but due to the lack of experimental data, it remains speculative. Some similar experiments as with TfaMu could have shed further insight. In retrospect, I feel that the title “... tail fibers with transient chaperone caps” was shaping higher expectations than were delivered.

We thank the reviewer for their assessment of our work and valuable feedback. As also mentioned in the discussion section, we acknowledge that a specific and confirmed role for these transient chaperone caps in relation to phage infectivity does require further experimentation. Nevertheless, we believe we have presented a comprehensive and sufficient set of observations that demonstrate (i) the transient nature by which these chaperones interact with their respective tail fibers, (ii) the structure of these chaperones and their relatedness to known intramolecular chaperones of other phage RBPs, and (iii) the difference in chaperone dependency of A5 and V22-like fibers to be functional and how this relates to their structural differences. As such, we are confident in the data we present to support our capping hypothesis, and we are motivated to continue this research, where we can fully explore the temporal aspects of chaperone interactions and their relevance to infection for future publication.

Regarding the strain panel used: We agree that the strain panel is too restrictive to make claims about phage host range. However, in this study specificity is not a relevant issue as we are not trying to identify and quantify the host range of phage A5. As such, the number of strains selected was based on the host-range analysis performed with the myophage V22 in our previous study (Gonzalez-Serrano 2020 et al.). Importantly, we are not making any claims about the receptor bound, or how broad these phages are. To avoid any confusion, we have changed line 166 from “*the three phages have extremely narrow host ranges*” to “*the three phages were not able to infect the other Alteromonas strains tested, with phages A5 and V22 shown to only infect A. mediterranea strains PT15 and PT11, respectively....*”

Regarding interaction analysis using SEC with/without salt: We agree with the reviewer that more advanced analysis would be preferable to complement the size exclusion chromatography data. However, while this would benefit a more robust understanding of the interaction between the chaperones and the fibers, it would not change the overall observation that the A5 chaperone is capable

of longer interaction with the tail fiber compared to the truncated V22 chaperone. This is the clear observation and while extra data could complement, it would not alter the original observation.

Regarding the comment “It also remains an unsolved question why the chaperone of A5 is not essential in spite of its stronger TF interactions.”: We agree with the reviewer. Further experimentation would be required to explore the temporal aspects of chaperone interactions and their relevance to infection.

Minor comments:

Line 123: three predicted RNA polymerase genes (since no experimental evidence is given)

Thank you. We have made this change (now line 124).

Figure 2: For phage V22 and phage P24, the old taxonomy is used in contrast to phage A5.

Thank you for bringing this to our attention. We have edited the main text and Figure 2 to accommodate for the updated taxonomy of bacteriophages. On line 160, this now reads: “*A5 is a schitovirus (of podoviral morphology), V22 is a myoalterovirus (of myoviral morphology), and P24 remains an unclassified Caudoviricetes (of siphoviral morphology)*”. We have also removed mention of taxonomic groups from Fig. 2. However, we retained our mention of viral morphology in panel B (and as described above) as it relates to the TEM images of these phages shown in panel C and provides context to our work when we describe our identification of “homologous host recognition modules used by morphologically distinct phages”.

Line 130: In fact, this is not surprising but frequently observed before for other phages.

We agree and have removed “surprisingly” from the text.

Figure 3: The phase contrast settings do not seem to be consistent across the different panels.

Due to differences in image brightness and cell visibility in the phase contrast images, the brightness and contrast of these micrographs were auto adjusted using ImageJ. This was only done to the phase contrast images to make the cells more visible. Importantly, none of the fluorescence images have been modified. If our work is accepted, we are happy to provide the original micrographs as source data, or if requested, use non-adjusted phase contrast images in Fig. 3A. We have also added the following statement to the materials and methods section on fluorescence microscopy (line 532): “*To improve visualization, the brightness and contrast were auto adjusted using ImageJ for the Phase Contrast micrographs only.*”

It is important to note that the microscopy data presented in Fig. 3A is only qualitative and provided us with a binary (yes/no) answer if the GFP-tagged proteins were bound to cells.

Line 264-265: ‘strongly suggests’ would be more correct than ‘strongly implies’

Agreed and changed.

Reviewer #2 (Remarks to the Author):

The manuscript of Gonzalez-Serrano et al utilizes in a sophisticated manner the modern bioinformatics tools to elucidate the functions of different components of bacteriophage tail fibers. The manuscript is very well written and enjoyable reading. The authors nicely demonstrate that host recognition modules of even distantly related phages are similar, and very likely spread via horizontal gene transfer. I don't identify any (severe) flaws in the manuscript.

Thank you for your review of our work and we are delighted to hear you found our analysis and presentation of the data engaging. We have made the necessary corrections to the text based on your suggestions below.

I have just a few minor points.

L114. Better: (coding for, e.g., integrases) Done

L414. Dialyzed Done

Figure 1 legend. ... colored based on detected molecular functions of individual gene products. Done

Sincerely

Mikael Skurnik

Reviewer #3 (Remarks to the Author):

In this well-written and enjoyable to read manuscript, Gonzalez-Serrano et al. study and compare the genomes of several Alteromonas phages and find highly homologous receptor-binding proteins between otherwise genetically and morphologically different bacteriophages.

They analyse the structures of the tail fibres and chaperones as predicted by AlphaFold-Multimer and find a more than passing resemblance with pyocin fibres and other bacteriophage receptor-binding proteins. In particular, a lectin-like fold containing head domain, trimeric knob domains and an alpha-beta chaperone fold, with in some cases an extended beta-hairpin loop that presumably wraps around the fibre.

This is a well-performed study, with sound methodology and no apparent flaws in data analysis.

They draw some very interesting but not completely new conclusions from the work:

- the conservation/horizontal transfer of host cell receptor binding units between very different phages, and for the biologically important Alteromonas phage is very important, although similar to other cases, as correctly referenced in the manuscript by the authors.
- the potential function of the chaperone not only in folding, but as a "cap" on the tail fibre to prevent premature binding to cell membranes of the host cell. I wonder if an experiment to further prove this function might be possible.

An interesting point is the mechanism of exchange, as discussed on lines 331-340. Would it be possible (now, or at some point in the future when even more phage and host sequences will be available), to discriminate between the ancient and more recent mechanism? Perhaps via some kind of genetic, sequence or structural signature? Presumably these events are too rare, but if the recent mechanism is true, could it be observed in the lab when infecting bacteria containing prophages with a suboptimal new phage - in a many-generation experiment?

We thank the reviewer for their kind comments on our work and for their suggestions for improving our investigations. We are confident in our data and analysis that these chaperones exist as transient fiber

caps based on the current data; however, we completely agree (as relayed to reviewer 1) that a defined biological function of these caps remains unknown and will be the subject of follow up experiments as suggested. We are highly interested in observing the transient interaction of the chaperones to the tail fibers of whole phage particles and observing how this affects phage infectivity. Given the high degree of similarity between the A5 and V22 tail fibers and chaperones it would also be interesting to perform infection experiments with a series of chaperone \pm fiber swapping between A5 and V22 to see how this could affect infectivity and host range. It still needs to be observed if the chaperone can re-attach once removed from the fiber or if this is an irreversible interaction as chaperone complementation assays could also be performed to measure whole phage infectivity. As we write on line 403 of the discussion: *“Further investigations are necessary to unravel how such a secondary capping function of these chaperones could mediate infectivity of whole phage particles as well as explain how these homologous host recognition module components can assemble into the three diverse infectious machineries used by these distantly related Alteromonas phages.”*

In sum, we believe we have presented the necessary evidence to support our capping hypothesis for these chaperones and will reserve the demonstration of its biological effect to future studies.

Regarding the mechanism of exchange: We appreciate the reviewer's suggestion to discriminate between the ancient and more recent mechanisms of exchange discussed in lines 336-345. It would indeed be valuable to explore this hypothesis with a larger number of phages and their sequenced genomes in the future. However, tracing the evolution of the host recognition module from its ancestral phages is challenging, as the evolutionary events that shaped the genomes of the phages studied in this research were unique. Conducting experimental evolution experiments using present-day phages may yield different results due to the potential for recombination and mutation events that generate different modular configurations. Nevertheless, it would be interesting to perform such a continuous evolution experiment and simply see which mechanism would be favored or provide the evolutionary advantage. This would also be interesting to analyze for non-*Alteromonas* phages carrying the A5/V22-like machinery that target clinically relevant pathogens to see if phage potency can be enhanced through domain duplication (A5) or truncation (V22). Regrettably, we do not currently have the capacity to perform such experiments in the lab.

Minor comments:

- I would avoid the abbreviation TF and just write tail fibre in full, some readers might confuse TF with for example transcription factor.

Thank you for this suggestion. We have exchanged TF to tail fiber throughout the article.

- perhaps include a suggestion that if salt partially prevents dissociation of the RBP-chaperone complex (gp8-gp9), the interactions between them are mainly hydrophobic? This might also be apparent from a structural model of the complex?

Thank you for the suggestion. While AlphaFold could predict the remarkable complexes formed by A5 and V22 fibers and chaperones (Fig. 6), one of the current limitations of AlphaFold and other structure prediction tools is the accuracy of protein-protein interactions. Upon closer inspection of the tentacle domains, they appear to project perpendicular to the body of the fiber and are not engaging the fiber body as was previously observed in crystal structures of the K1F and T5 RBPs where the IMC domain tentacles are situated along the grooves of the RBP surface and form different interactions. As such, it is difficult to say with certainty the type of interaction formed between the tentacles and the fiber body for A5 and V22. Nevertheless, we agree with the reviewer that it is important to improve our explanation of the tentacle composition and its possible mode of interaction. As such, in the discussion (line 356) we have added the following: *“The A5 and V22 tentacles also contain a high proportion of polar*

residues, similar to those found in the tentacles of the KIF and T5 IMCs. This similarity would facilitate a comparable network of hydrophilic interactions with the central RBP, as previously described 43,44. However, despite the A5 and V22 complexes being predicted with high confidence, the tentacles are modeled as perpendicular extensions to the fiber and are too distant to suggest a network of interactions between the chaperone and fiber tip. Nevertheless, a clear interaction between both fibers and respective chaperones was identified as hydrogen bonding between β -strands of both proteins.”

Reviewers' Comments:

Reviewer #1:

Remarks to the Author:

Line 317: the authors forgot to change Myovirus to myoalterovirus here.

Reviewer #3:

Remarks to the Author:

The authors have incorporated all the suggested text edits and edited the manuscript to better make their point, but no additional experimental evidence was provided. To me, as a phage researcher, the manuscript is very interesting. It is up to the Editor to decide if it is sufficiently interesting for a more general readership.

A host recognition module shared among distant *Alteromonas* bacteriophage families use tail fibers exhibiting properties of transient chaperone “caps”

- Response to Reviewers II -

Reviewer #1 (Remarks to the Author):

Line 317: the authors forgot to change Myovirus to myoalterovirus here.

We appreciate the reviewer's positive remarks on our work through both of their thorough reviews. Thank you also for spotting this mistake in the main text. We have corrected the designations for V22 and P24 in this section based on recent taxonomic reclassifications.

Revised line 317: Schitovirus A5, myoalterovirus V22, and unclassified Caudoviricetes (siphoviral morphology) P24 share a common host recognition module...

Reviewer #3 (Remarks to the Author):

The authors have incorporated all the suggested text edits and edited the manuscript to better make their point, but no additional experimental evidence was provided. To me, as a phage researcher, the manuscript is very interesting. It is up to the Editor to decide if it is sufficiently interesting for a more general readership.

Our sincere thanks to the reviewer for their encouraging feedback after revisiting our work.